# YAP1 Expression in HR+HER2− Breast Cancer: 21-Gene Recurrence Score Analysis and Public Dataset Validation

**DOI:** 10.3390/cancers15205034

**Published:** 2023-10-18

**Authors:** Inho Park, Yangkyu Lee, Jee Hung Kim, Soong June Bae, Sung Gwe Ahn, Joon Jeong, Yoon Jin Cha

**Affiliations:** 1Department of Pathology, Gangnam Severance Hospital, Yonsei University College of Medicine, Seoul 06273, Republic of Korea; ihpark20@yuhs.ac (I.P.); maytangerinetree@yuhs.ac (Y.L.); 2Center for Precision Medicine, Gangnam Severance Hospital, Yonsei University College of Medicine, Seoul 06273, Republic of Korea; 3Institute of Breast Cancer Precision Medicine, Yonsei University College of Medicine, Seoul 06273, Republic of Korea; ok8504@yuhs.ac (J.H.K.); mission815815@yuhs.ac (S.J.B.); asg2004@yuhs.ac (S.G.A.); gsjjoon@yuhs.ac (J.J.); 4Division of Medical Oncology, Department of Internal Medicine, Yonsei University College of Medicine, Seoul 06273, Republic of Korea; 5Department of Surgery, Gangnam Severance Hospital, Yonsei University College of Medicine, Seoul 06273, Republic of Korea

**Keywords:** breast neoplasms, Yes-associated protein 1, prognosis, receptors, estrogen, pathology

## Abstract

**Simple Summary:**

YAP1, a downstream transcription factor of the Hippo pathway, is regarded as an oncogene in various solid tumors. This study explores the relationship between YAP1 expression and the risk score from the Oncotype Dx test in patients with hormone-receptor-positive, HER2-negative (HR+HER2−) breast cancer. In a retrospective review of 401 patients using YAP1 nuclear localization via immunohistochemical staining and clinicopathologic analysis, high-YAP1 expression significantly correlated with a lower risk score. In a public dataset analysis, elevated YAP1 mRNA expression was associated with better clinical outcomes, particularly in ER-positive patients. In summary, YAP1 could serve as a prognostic marker as well as potential therapeutic target in HR+HER2− breast cancer patients.

**Abstract:**

Background: YAP1, an oncogene in numerous cancers, is a downstream transcription factor of the Hippo pathway. This study focuses on its relationship with the Oncotype Dx (ODX) test risk score (RS) in patients with hormone-receptor-positive, HER2-negative (HR+HER2−) breast cancer. Methods: We retrospectively analyzed 401 HR+HER2− breast cancer patients from Gangnam Severance Hospital who underwent ODX tests (May 2014–April 2020). YAP1 nuclear localization was evaluated via immunohistochemical staining and its clinical correlation with clinicopathological parameters, including RS, was analyzed. Public datasets TCGA-BRCA and METABRIC validated clinical outcomes. Results: YAP1 expression negatively correlated with ODX RS (OR 0.373, *p* = 0.002). Elevated YAP1 mRNA levels corresponded to better clinical outcomes, specifically in ER-positive patients, with significant results in METABRIC and TCGA-BRCA datasets (*p* < 0.0001 OS in METABRIC, *p* = 0.00085 RFS in METABRIC, *p* = 0.040 DFS in TCGA-BRCA). In subsets with varying ESR1 mRNA expression and pronounced YAP1 expression, superior survival outcomes were consistently observed. Conclusion: YAP1 may be a valuable prognostic marker and potential therapeutic target in HR+HER2− breast cancer patients.

## 1. Introduction

Breast cancer is the most commonly diagnosed cancer in women worldwide [1,2]. Clinically, breast cancer is subdivided into three subtypes based on the immunohistochemistry (IHC) status of hormone receptors’ (HRs) estrogen receptor (ER) and progesterone receptor (PR) and human epidermal growth factor receptor 2 (HER2). Among them, HR-positive-HER2-negative (HR+HER2−) breast cancer is the most predominant subtype, accounting for nearly 70% of breast cancer [2]. HR+HER2− breast cancer is generally less aggressive and has anti-hormonal therapeutic options such as tamoxifen, with a relatively superior prognosis compared to other subtypes [2]. Oncotype DX 21-gene Breast Cancer Recurrence Score ^®^ assay (ODX RS) (Genomic Health; Redwood City, CA, USA) is a precise genomic test based on the gene expression level of 21 genes in breast cancer tumor tissue [3], provided the recurrence score (RS) is (0–100). A high RS (usually >26) indicates that the patient belongs to the high-risk group and might benefit from chemotherapy [3].

Yes-associated protein 1 (YAP1) and transcriptional coactivator with PDZ-binding motif (TAZ) are downstream transcription factors of the Hippo pathway. When the Hippo pathway is disrupted, unphosphorylated YAP1/TAZ moves into the nucleus of the cells and, together with the TEA domain transcription factor (TEAD), upregulate the transcription of multiple genes involved in cell proliferation, apoptosis, and survival [4]. YAP1, together with TAZ, has been considered an oncogene as overexpression or activation of YAP1 correlated with poor prognosis with cancer of variable organs, including ovary, [5,6] lung, [7] esophagus, [8] colon, [9], and pancreas [10]. In breast cancer, there have been controversies on the role of YAP1 in tumor biology. YAP1 was reported as tumor suppressor in some studies [11,12], whereas it was correlated with poor prognosis in others [13,14,15].

In a previous study, we showed that YAP1 nuclear expression was increased along with tumor stiffness measured by shear-wave elastography in HR+HER2− breast cancer [16]. Tumor stiffness is derived from tumor cell invasion beyond the basement membrane and the production of collagen by cancer-associated fibroblasts in the extracellular matrix of the tumor microenvironment, which is also known as desmoplastic reaction [17]. High tumor stiffness in breast cancer is an indicator of aggressive histologic features and differentiates high-risk patient groups [18,19,20,21]. Based on the positive correlation between YAP1 expression and tumor stiffness in HR+HER2− breast cancer, we further aimed to evaluate whether YAP1 expression correlates with ODX RS. As most of the patients with HR+HER2− breast cancer have excellent 5-year survival with very rare events to analyze in the limited period, we expected that ODX RS could serve as a surrogate marker for a prognosis for patients with HR+HER2− breast cancer.

## 2. Materials and Methods

This study was approved by the Institutional Review Board of Gangnam Severance Hospital (3-2022-0119) and adhered to the clinical practice guidelines of the Declaration of Helsinki (2013 amendment). Informed consent was waived for all patients due to the retrospective design of this study.

### 2.1. Patients

We retrospectively selected patients who underwent upfront curative surgery followed by adjuvant treatments for breast cancer at Gangnam Severance Hospital in Seoul, Korea, from May 2014 to April 2020. The clinical and pathological data of the patients were obtained by reviewing the electronic medical records (EMRs). Stages were determined according to the 8th edition of the American Joint Committee on Cancer staging system. Clinicopathologic parameters evaluated in each case from EMR included patient age at initial diagnosis, menopausal status, tumor size, histologic grade (HG) based on the Nottingham grading system [22], lymphovascular invasion (LVI), lymph node metastasis, tumor recurrence, distant metastasis, patient survival, and ODX RS.

The inclusion and exclusion criteria were as follows:Inclusion criteria:patients aged ≥20 years;invasive breast cancer confirmed by pathological diagnosis;available ODX RS;ER- and/or PR-positive and HER2-negative cancer.Exclusion criteria:any other carcinoma in situ;other cancer histories (except for thyroid cancer);inaccessible electronic medical records;received neoadjuvant chemotherapy (NAC).

### 2.2. Oncotype Dx^®^ Assays

The ODX assay was performed using RNA extracted from formalin-fixed, paraffin-embedded (FFPE) tissue and supplied by Genomic Health (Redwood City, CA, USA). RNA was extracted from unstained sections containing sufficient invasive breast cancer tissue of appropriate quality. Patients with an RS of 26 or higher were assigned to a high-risk group based on the TAILORX trial [23].

### 2.3. Pathologic Review of Breast Cancer Slides

#### 2.3.1. Histologic Evaluation of the Tumor–Stroma Ratio (TSR) and Tumor-Infiltrating Lymphocytes (TILs)

Histology slides of patients were reviewed by two breast pathologists (Y.K. and Y.J.C.). The tumor–stroma ratio (TSR) is defined as tumor cellularity relative to the surrounding stroma in the overall tumor bed [16,24]. The TSR assessment was conducted using scoring percentages in 10% increments. For statistical analysis, cases with ≤50% TSR were assigned to the stroma-high group, and those with >50% TSR were assigned to the stroma-low group. The tumor-infiltrating lymphocyte (TIL) level was concurrently evaluated according to the guidelines suggested by the International TIL Working Group [25]. Except for polymorphonuclear leukocytes, other mononuclear cells, including lymphocytes and plasma cells, were counted. For statistical analysis, a 10% cutoff was applied to separate patients into low-TIL (<10%) and high-TIL (≥10%) groups.

#### 2.3.2. IHC for Clinical Subtype

Nuclear staining values of 1% or higher were considered positive for ER (clone 6F11; dilution 1:200; Leica Biosystems, Wetzlar, Germany) and PR (clone 16; dilution 1:500; Leica Biosystems, Wetzlar, Germany) [26]. HER2 (clone 4B5; dilution 1:5; Ventana Medical System, Oro Valley, AZ, USA) staining was performed according to the 2018 American Society of Clinical Oncology/College of American Pathologists [27]. Only samples with strong and circumferential membranous HER2 immunoreactivity (3+) were considered positive, whereas those with 0 or 1+ HER2 staining were considered negative. Cases with equivocal HER2 expression (2+) were further evaluated for HER2 gene amplification via silver in situ hybridization (SISH). Positive nuclear Ki67 (clone MIB; dilution 1:1000; Abcam, Cambridge, UK) staining was assessed based on the percentage of positive tumor cells, defined as the Ki67 labelling index (LI). ER- and/or PR-positive and HER2-negative cases were selected for this study.

### 2.4. Tissue Microarray (TMA) Construction

Hematoxylin and eosin (H&E)-stained slides from the resected breast cancer specimens were examined and representative areas marked. The matched tissue cores (2 mm) were extracted from FFPE tumor blocks and placed into 5 × 10 recipient TMA blocks. Each tissue core was assigned a unique TMA location number that was linked to a database containing other clinicopathologic data.

### 2.5. YAP1 IHC and Interpretation

Briefly, 3 µm thick tissue sections were cut from the FFPE tissue block of the TMA blocks. After deparaffinization and rehydration with graded xylene and alcohol solutions, IHC was performed using a Ventana Discovery XT Automated Slide Stainer (Ventana Medical System, Tucson, AZ, USA). Cell conditioning 1 (CC1) buffer (citrate buffer, pH 6.0; Ventana Medical System) was used for antigen retrieval. Whole tissue slides were stained with an anti-YAP1 antibody (clone 63.7; dilution 1:200; Santa Cruz Biotechnology, Dallas, TX, USA). After staining, nuclear YAP1 expression was assessed by two breast pathologists (YL and YJC; 400× magnification). Nuclear staining was evaluated using the H-score, which was obtained by multiplying the staining intensity (0, 1, 2, or 3) by the percentage of stained area (%). The myoepithelial cells’ nuclear staining intensity was assigned a value of moderate intensity and used as an internal control. Weaker and stronger signals were assigned a value of weak and strong intensities, respectively. Negative or weak nuclear staining were categorized as low expression, while moderate or strong nuclear staining were grouped as high expression (Figure 1). The IHC results were interpreted blindly, without any information regarding clinical parameters or outcomes.

### 2.6. Public Dataset Analysis

We investigated the relationship between YAP1 expression and survival outcomes, specifically overall survival (OS) and event free survival (EFS), across various subgroups within the two publicly available datasets: The Cancer Genome Atlas Breast Invasive Carcinoma (TCGA-BRCA) and The Molecular Taxonomy of Breast Cancer International Consortium (METABRIC). The clinical and gene expression data were obtained from the cBioPortal (https://www.cbioportal.org, accessed on 9 May 2023). In our analysis, we considered disease-free survival (DFS) from the TCGA-BRCA dataset and relapse-free survival (RFS) from METABRIC as EFS. Subgroups analyzed included IHC based ER-positive (ER+)/negative (ER−) and PAM50-based molecular subtypes. We utilized gene expression data in the form of z-scores relative to normal samples, which were obtained from the cBioPortal.

Within each subgroup, we evaluated the differences in survival outcomes between the dichotomized samples for every potential YAP1 expression threshold, covering the range from the 10th to the 90th percentile of YAP1 expression within the tumor samples. The optimal dichotomization was determined based on the *p*-value obtained from the log-rank survival difference test. Survival outcome analysis based on YAP1 expression was conducted in patients with varying levels of ESR1 expression, particularly those with ESR1 levels in the upper 20-, 40-, 60-, and 80 percentiles.

### 2.7. Statistical Analysis

The continuous variables between the two groups were compared using the Student’s *t*-test or the Mann–Whitney test. The categorical variables were compared by using the Chi-square test or Fisher’s exact test. Survival curves were obtained using the Kaplan–Meier method and two-group comparisons were made using the log-rank test. Univariate and multivariate regression analyses were conducted to identify the significant parameters. Statistical analyses were performed using SPSS version 24 (IBM co Chicago, IL, USA) and R software (https://www.r-projet.org; version 4.3.0, accessed on 9 May 2023). The threshold for statistical significance was set at *p* < 0.05, with a 95% confidence interval (CI) not including 1.

## 3. Results

### 3.1. Basal Characteristics of the Study Population

Data from a total of 401 female patients were evaluated and summarized in Appendix A. The median age of patients was 49 years (range 25–81 years). Patients with premenopausal status were larger in number (*n* = 235, 58.6%) than those with menopausal status (*n* = 155, 38.7%). Most patients were in the early stage of the tumor (pT1, *n* = 246, 61.3%; pT2, *n* = 154, 38.4%; pT3, *n* = 1, 0.2%) with a mean tumor size of 1.9 cm, and invasive ductal carcinoma was the most predominant histologic subtype (91.0%). The mean RS of the ODX assay was 18.1, and applying the RS 26 cutoff, 340 (84.8%) and 61 (15.2%) patients were assigned to low- and high-risk groups, respectively. Regarding pathologic parameters, histologic grade (HG) II was the most common (*n* = 311, 77.6%), followed by HG I (*n* = 63, 15.7%) and HG III (*n* = 27, 6.7%). LVI was detected in 122 cases (30.4%), and lymph node metastasis was present in 96 (23.9%) cases. Overall, the mean TIL level was 11.2%, and using a 10% cutoff, 302 (75.3%) and 99 (24.7%) cases were subgrouped as low- and high-TIL groups, respectively. The mean TSR was 71.0%, and using a 50% cutoff, 344 (86.0%) and 56 (14.0%) cases were classified into stroma-low and stroma-high group, respectively. With dichotomized YAP1 scoring, 139 (34.7%) and 262 (65.3%) cases corresponded to low- and high-YAP1 groups, respectively.

### 3.2. Comparison of Clinicopathologic Factors Based on YAP1 Expression and ODX RS

We investigated the characteristics of each group based on the YAP1 expression and ODX RS. First, we compared the clinicopathologic parameters, including ODX RS, between low- and high-YAP1 groups (Table 1). The low-YAP1 group showed a significantly higher ODX RS (20.4 ± 11.6 vs. 16.8 ± 7.2, *p* = 0.001), and a had significantly larger proportion of patients in the high-risk group (25.9% vs. 9.5%, *p* < 0.001) than those in high-YAP1 group. The distribution of HG differed; specifically, HG III was significantly frequent, four times higher, in the low-YAP1 group than in the high-YAP1 group (12.9% vs. 3.4%, *p* = 0.001). The proliferation index was significantly higher in the low-YAP1 group (13.8 ± 9.5 vs. 10.9 ± 10.9, *p* = 0.007). Conversely, there was no difference in menopausal status, tumor size, lymph node metastasis, LVI, TIL, or TSR based on the YAP1 expression.

Second, we performed the *t*-test using the ODX risk group (Table 2). The high-risk group had a considerably larger number of patients with menopause than the low-risk group (60.7% vs. 34.7%, *p* = 0.001), as well as aggressive histologic features, including HG III (*p* < 0.001) and higher Ki67 LI (*p* < 0.001). However, lymph node metastasis (6.6% in high-risk group vs. 27.1% in low-risk group, *p* = 0.001) and LVI (19.7% in high-risk group vs. 32.4% in the low-risk group, *p* = 0.047) were significantly more frequent in the low-risk group. No significant difference was found in tumor size, TIL, or TSR in the RS group.

### 3.3. Correlation of YAP1 Expression and ODX RS Using Regression Analysis

In univariate linear regression analysis with ODX RS as a continuous variable, significantly related parameters with higher ODX RS were menopause status (*p* = 0.021), higher TIL level (*p* = 0.006), higher Ki67 LI (*p* < 0.001), higher HG (*p* < 0.001), absence of lymph node metastasis (*p* < 0.001), and low YAP1 expression (*p* < 0.001) (Table 3). Multivariate linear regression analysis revealed menopause status (*p* = 0.047) and higher Ki67 LI (*p* < 0.001) as positively correlated factors with ODX RS, whereas lymph node metastasis (*p* = 0.003) and YAP1 expression (*p* = 0.002) showed a significant negative correlation with ODX RS (Table 4).

Additionally, logistic regression analysis was performed to find the significant parameters of the ODX high-risk group (Table 4). In univariate analysis, menopause and HG III were significantly associated with RS ≥ 26 (menopause: odds ratio [OR] 2.890, 95% CI 1.639–5.096, *p* < 0.001; HG III: OR 10.14, 95% CI 2.846–36.136, *p* < 0.001). High-TIL and high Ki67 LI also showed significant association with ODX high-risk group, with an OR of 1.017 and 1.078, respectively. Lymph node metastasis and high YAP1 expression showed significant association with the ODX low-risk group of RS < 26 (lymph node metastasis: OR 0.189, 95%CI 0.067–0.536, *p* = 0.002; high YAP1: OR 0.302, 95% CI 0.172–0.528, *p* < 0.001). In multivariate analysis, menopause (OR 2.897, 95% CI 1.538–5.454, *p* < 0.001), HG III (OR 4.625, 95% CI 1.070–19.990, *p* = 0.040), and high Ki67 LI (OR 1.062, 95% CI 1.035–1.090, *p* < 0.001) showed significant association with RS ≥ 26. With lymph node metastasis (OR 0.270, 95% CI 0.091–0.801, *p* = 0.018), high YAP1 expression remained an independently significant parameter with ODX low risk, RS ≤26 (OR 0.373, 95% CI 0.198–0.703, *p* = 0.002).

### 3.4. Validation of the Prognostic Effect of YAP1 Expression in Public Datasets

In METABRIC and TCGA-BRCA datasets, high YAP1 expression showed superior clinical outcomes in ER+ patients, whereas no significant difference was found in ER-negative patients (ER+ vs. ER−: *p* < 0.0001 vs. *p* = 0.130, OS in METABRIC; *p* = 0.00085 vs. *p* = 0.260, RFS in METABRIC; *p* = 0.040 vs. *p* = 0.260, DFS in TCGA-BRCA) (Figure 2). In the METABRIC dataset, the prognostic effect of YAP1 expression level on ER+ patients was only significant in patients of postmenopausal status (OS, *p* = 0.012; RFS, *p* = 0.00098).

Based on the molecular subtype, high YAP1 expression demonstrated superior OS in the luminal A subtype (*p* < 0.0001), while exhibiting significantly improved RFS specifically in the luminal B subtype (*p* = 0.020) and a tendency towards better RFS in the luminal A subtype (*p* = 0.089) within the METABRIC dataset. Conversely, high YAP1 expression in HER2 molecular subtype showed significantly worse OS (*p* = 0.0042) and a tendency of worse RFS (*p* = 0.059). In the TCGA-BRCA dataset, high YAP1 expression was significantly associated exclusively with the luminal A subtype (*p* = 0.0018). Patients with high YAP1 expression tended to have worse DFS in the luminal B subtype (*p* = 0.069) (Figure 3).

With different ESR1 expression levels in the METABRIC dataset, patients having ESR1 level in the upper 80, 60, and 40 percentiles, high YAP1 expression consistently demonstrated superior OS (*p* < 0.0001, *p* < 0.0001, and *p* = 0.003, respectively) and RFS (*p* = 0.0031, *p* < 0.0001, and *p* = 0.00013, respectively). Conversely, for patients in the upper 20 percentiles of ESR1 expression level, high YAP1 expression still showed better RFS (*p* = 0.024) but not significant OS (*p* = 0.210). In the TCGA-BRCA dataset, patients within upper 60 and 40 ESR1 expression percentiles exhibited superior DFS with high YAP1 expression (*p* = 0.013 and *p* = 0.011, respectively), whereas those in the upper 80 and 20 percentiles of ESR1 levels only showed a tendency towards better DFS (*p* = 0.060 and *p* = 0.076, respectively) (Figure 4).

## 4. Discussion

In this study, YAP1 activation was clearly correlated with lower RS and independently predicted the lower RS. This result implies that YAP1 functions like a tumor-suppressor in HR+HER− breast cancer, which was further supported by the analytic result of public datasets. Patients with HR+HER2− breast cancer have the most favorable prognosis, with 94.8% of 5-year relative survival rate [28]. However, drug resistance to hormone therapy develops over time, and at the advanced stage, the relative survival rate of patients with HR+HER2 breast cancer is decreased, even worse than that of patients with HER2+ breast cancer [28].

ODX is a multigene test that predicts the benefit of chemotherapy in patients with HR+HER2− breast cancer. In low-grade breast cancer cases, chemotherapy administration has no therapeutic benefit that overcomes the side effects of treatment. ODX RS ≥ 26 is the general cutoff for patients at high risk that might benefit from additional chemotherapy. Since surgically treated HR+HER2− breast cancer has nearly a 100% 5-year survival, we used ODX RS as a surrogate for tumor aggressiveness in this study.

YAP1 has been described as an oncogene in diverse organs [5,6,8,9,10]. However, in breast cancer, controversies regarding the role of YAP1 in tumor biology exist [11,15,29,30,31]. In our previous study with triple-negative breast cancer (TNBC), YAP1 activation was correlated with a poor clinical outcome [13]. As multiple previous studies did not specifically examine the subcellular localization of YAP1 expression (nucleus or cytoplasm) [11,29,32], we conducted a comprehensive YAP1 immunohistochemistry (IHC) interpretation to confirm its nuclear localization and determine YAP1 activation. Another previous study of our group showed a correlation between YAP1 expression and tumor stiffness in HR+HER2− breast cancer tissue [16]. As tumor stiffness in breast cancer is associated with aggressive features affecting prognosis [18,19,20], we expected a correlation between high YAP1 expression and high ODX RS in this study.

Regarding ODX RS, high RS showed significant correlation with high HG and high Ki67 LI, which are classical parameters of tumor aggressiveness. What was unexpected was the inverse correlation between RS and LVI or lymph node metastasis; however, this appeared to be an intrinsic limitation of ODX assay, as it focused on the proliferation and invasion of primary breast cancer cells [3]. Several studies have highlighted the limited reliability of ODX assay in nodal burden prediction [33,34,35].

In this study, YAP1 expression showed a different pattern compared to our previous study with TNBC [13]. In HR+HER2− breast cancer, high YAP1 expression was associated with low HG, low Ki67 LI, and low ODX RS. This favorable prognostic impact of YAP1 expression was also seen in the public datasets analyses, which showed significant superior survival in high-YAP1 group of IHC-defined ER+ breast cancer. Recently, several studies regarding the Hippo signaling pathway and ERα regulation suggested the inhibitory role of YAP1 on ER+ breast cancer growth [36,37,38]. In ER+ breast cancer cell lines, LATS1/2, upstream inhibitors of YAP1, are required to maintain ER+ cancer cell growth while little effect was observed in ER− cancer cells [38]. Further study showed that YAP1, together with TEAD, targets VGLL3, which recruits NCOR2 and represses ESR1 transcription [37]. YAP1 physically interrupts the ERα/TEAD interaction by competing with ERα in ER+ breast cancer cells [36]. In in vitro analysis, ERα is dissociated from its target promoters/enhancers by YAP1, which results in ERα degradation and subsequent ESR1 gene downregulation [36]. Collectively, YAP1 acts differently in ER+ breast cancer cells by inhibiting the downstream signaling pathway of the ESR1 gene, which is crucial for ER+ tumor growth. Specifically, the favorable prognosis of high-YAP1 tumor in incremental ESR1 expression supports the inhibitory role of YAP1 on ESR1 signaling. However, in tumors of ESR1-level upper 20 percentiles, YAP1 seemed to have less effect on the favorable clinical outcome. Those high-ESR1 expression tumors are exclusively composed of a comparable number of luminal A and luminal B tumors (Appendix A). A luminal B tumor is an aggressive and proliferative subtype, which might not be solely dependent on ESR1 signaling. In addition, YAP1 might not be sufficient to inhibit the overwhelmingly expressed ESR1.

So far, there have been few studies reporting associations between YAP1 expression and favorable outcome in patients with luminal breast cancer [30,32]. Additionally, these studies had limitations in interpreting YAP1 subcellular localization or were conducted on small luminal cohorts. Therefore, the present study might be the first large-cohort study specifically comprised of HR+HER2− breast cancer with precise YAP1 IHC interpretation and validation of its clinical impact using public datasets. Also, we used human breast tissues and observed significant correlations between YAP1 expression and ODX RS, which confirmed previous in vitro studies [36,37,38], and clarified the role of YAP1 in breast cancer.

In the disease course of HR+HER2− breast cancer, the development of tamoxifen-resistance is an important clinical issue, which cannot be predicted with an ODX assay. The regulation of YAP1 expression could help in overcoming tamoxifen resistance [36,37,39,40]. The mechanism of endocrine resistance in ER+ breast cancer includes the loss of ERα expression, the mutation of the ESR1 gene, and the activation of alternative signaling pathways, such as HER2 [41]. In cases with lost ERα expression, YAP1 inhibition could restore the ERα expression that sensitizes the tamoxifen-resistant luminal cancer cell line [40]. Conversely, if mutant ERα causes tamoxifen resistance, the inhibition of the upstream molecules of YAP1, MST1/2, or LATS1/2 induce YAP1 activation, which could repress the downstream signaling of the ESR1 gene and overcome the resistance [36,37,39]. Notably, high YAP1 expression appeared to be more beneficial to postmenopausal than to premenopausal patients in this study. This could be explained by the lower estrogen levels in postmenopausal patients, which may have a similar effect to tamoxifen or facilitate YAP1 interference with ESR1 signaling more easily. The modulation of YAP1 activity could help to manage the long-term treatment plan in HR+HER2− breast cancer.

There are several limitations and future tasks to address. First, our cohort had a relatively short follow-up period with all alive patients and with only four recurrent cases. This was because we aimed to focus on YAP1 expression and ODX as sufficient follow-up data were unavailable. Although ODX RS showed a significant correlation with YAP1 expression, it only could provide the likelihood of risk. Second, our YAP1 activation was defined based on the IHC results of TMA. As YAP1 is heterogeneously expressed across the tumor, we might have missed the YAP1-positive area in low-YAP1 tumors. To compensate for these limitations, we validated our results with public datasets with larger numbers of patients with mRNA expression data; however, a more refined large dataset should be further validated combined with cancer stage, menopausal status, and tamoxifen treatment and resistance. Our study primarily concentrated on HR+HER2− breast cancer; however, the relevance of HER2, especially in contexts like ductal carcinoma in situ (DCIS), cannot be understated [42]. Furthermore, although our investigations have ascertained the detrimental prognostic implications of YAP1 in TNBC, the precise association between YAP1 expression and HER2 overexpression remains to be comprehensively elucidated. This represents a significant gap in the current body of knowledge and warrants rigorous examination in future studies. Lastly, the different role of YAP1 as a mechanotransducer in HR+HER2− breast cancer should be further elucidated in the context of clinical implication. As we observed in our previous study, YAP1 expression increased along with tumor stiffness in HR+HER2− breast cancer [16]. We could not analyze the association between YAP1, tumor stiffness, and clinical implication because of different patient cohorts. However, retrospective clinicopathological validation with the genomic dataset of advanced-stage HR+HER2− breast cancer cases might clarify the conundrum.

## 5. Conclusions

In conclusion, YAP1 acts as a tumor suppressor in HR+HER2− breast cancer. Interaction of YAP1 and ESR1 in nuclei is a unique feature of HR+HER2− breast cancer, and thus YAP1 could serve as a prognostic marker as well as a therapeutic target in patients with HR+HER2− breast cancer.

## Figures and Tables

**Figure 1 cancers-15-05034-f001:**
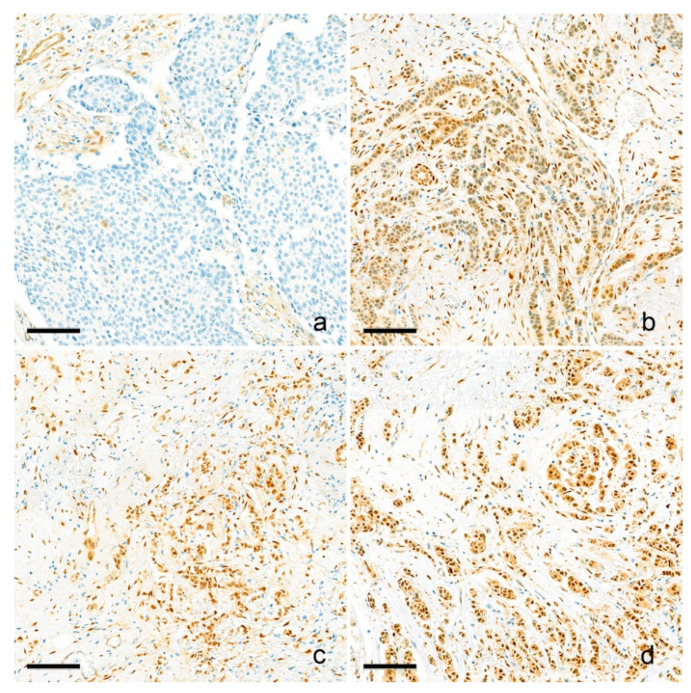
Representative pictures of YAP1 immunohistochemistry. Nuclear YAP1 expression is evaluated and scored as negative (no expression) (**a**), weak (**b**), moderate (**c**), and strong (**d**) nuclear expression. Negative/weak expressions are considered as low-YAP1 expression, and moderate/strong expressions are considered high YAP1 expression. Scale bar: 100 μm.

**Figure 2 cancers-15-05034-f002:**
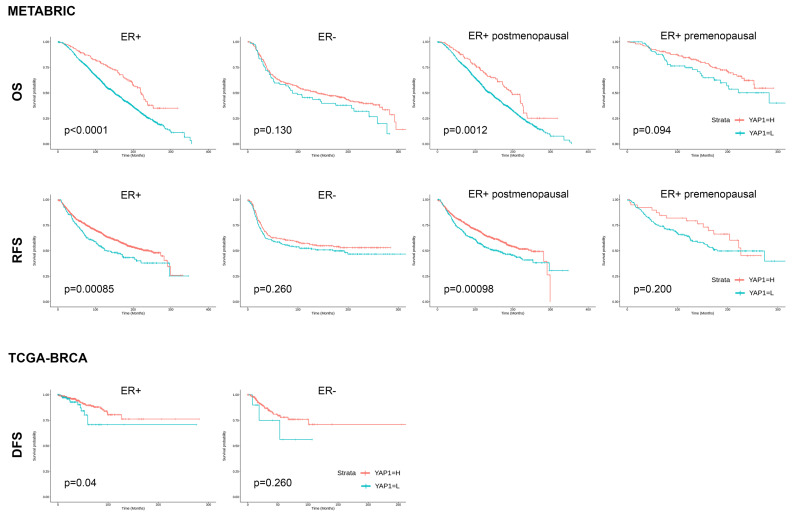
Clinical outcome of ER-positive and ER-negative breast cancer based on YAP1 expression.

**Figure 3 cancers-15-05034-f003:**
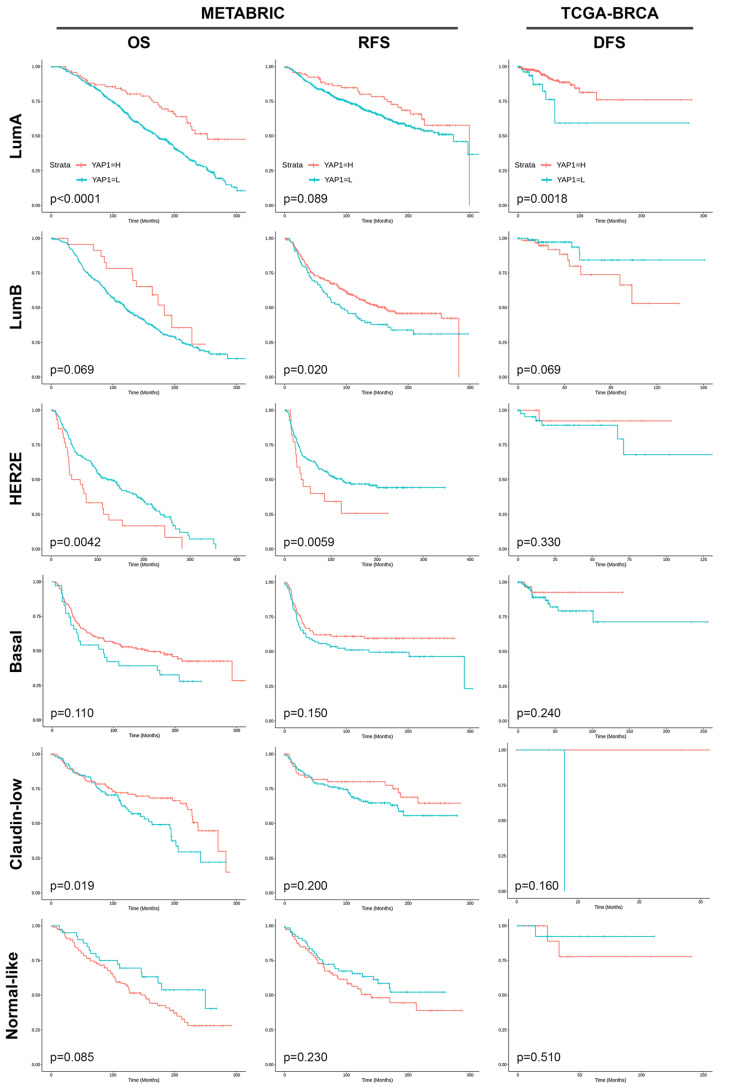
Clinical outcome of different molecular subtypes of breast cancer based on YAP1 expression.

**Figure 4 cancers-15-05034-f004:**
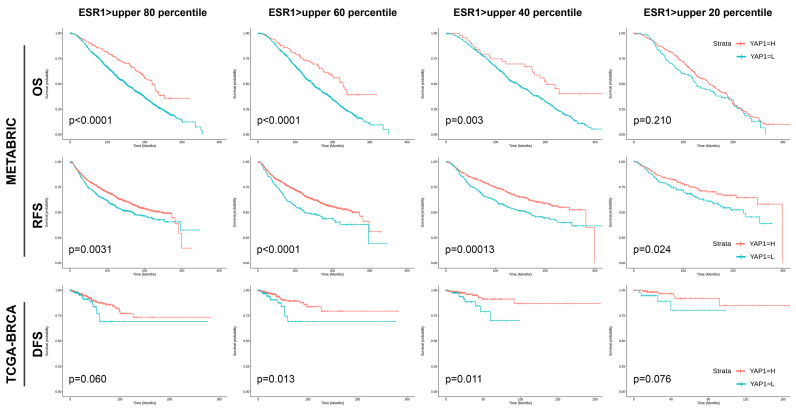
Clinical outcomes of patients with different ESR1 expression levels based on YAP1 expression.

**Table 1 cancers-15-05034-t001:** Comparison of clinicopathologic parameters based on the YAP1 expression.

Parameters	YAP1-Low (*n* = 139)	YAP1-High (*n* = 262)	*p*-Value
Menopausal Status, *n* (%)			0.343
Premenopausal	75 (54.0)	160 (61.1)	
Menopause	59 (42.4)	96 (36.6)	
Not assessable	5 (3.6)	6 (2.3)	
Oncotype Dx RS (mean ± SD)	20.4 ± 11.6	16.8 ± 7.2	0.001
ODX risk group, *n* (%)			<0.001
Low-risk (<26)	103 (74.1)	237 (90.5)	
High-risk (≥26)	36 (25.9)	25 (9.5)	
Histologic grade, *n* (%)			0.001
I	17 (12.2)	46 (17.6)	
II	104 (74.8)	207 (79.0)	
III	18 (12.9)	9 (3.4)	
Tumor size, cm (mean ± SD)	2.0 ± 0.8	2.0 ± 0.8	0.405
Lymph node metastasis, *n* (%)			0.624
Absent	108 (77.7)	197 (75.2)	
Present	31 (22.3)	65 (24.8)	
Lympovascular invasion, *n* (%)			1.000
Absent	97 (69.8)	182 (69.5)	
Present	42 (30.2)	80 (30.5)	
TIL level, % (mean ± SD)	11.6 ± 17.3	11.0 ± 11.4	0.693
TIL group, *n* (%)			0.903
Low-TIL (≤10%)	104 (74.8)	198 (75.6)	
High-TIL (>10%)	35 (25.2)	64 (24.4)	
TSR, % (mean ± SD)	71.9 ± 15.7	70.5 ± 1.2	0.454
TSR group, *n* (%)			0.226
Stroma-low (TSR >50%)	124 (89.2)	221 (84.4)	
Stroma-high (TSR ≤50%)	15 (10.8)	41 (15.6)	
Ki67 LI, % (mean ± SD)	13.8 ± 9.5	10.9 ± 10.9	0.007

YAP1, Yes-associated protein 1; RS, risk score; SD, standard deviation; ODX, oncotype Dx; TIL, tumor-infiltrating lymphocyte; TSR, tumor–stroma ratio; LI, labelling index.

**Table 2 cancers-15-05034-t002:** Comparison of clinicopathologic parameters based on the risk score.

Parameters	High-Risk (RS ≥ 26) (*n* = 61)	Low-Risk (RS < 26) (*n* = 340)	*p*-Value
Menopausal Status, *n* (%)			0.001
Premenopausal	23 (37.7)	212 (62.4)	
Menopause	37 (60.7)	118 (34.7)	
Not assessable	1 (1.6)	10 (2.9)	
YAP1 expression, *n* (%)			<0.001
Low-YAP1	36 (59.0)	103 (30.3)	
High-YAP1	25 (41.0)	237 (69.7)	
Histologic grade, *n* (%)			<0.001
I	4 (6.6)	59 (17.4)	
II	46 (75.4)	265 (77.9)	
III	11 (18.0)	16 (4.7)	
Tumor size, cm (mean ± SD)	2.0 ± 0.7	1.9 ± 0.8	0.919
Lymph node metastasis, *n* (%)			0.001
Absent	57 (93.4)	248 (72.9)	
Present	4 (6.6)	92 (27.1)	
Lymphovascular invasion, *n* (%)			0.047
Absent	49 (80.3)	230 (67.6)	
Present	12 (19.7)	110 (32.4)	
TIL, % (mean ± SD)	14.5 ± 20.7	10.6 ± 12.0	0.157
TIL group, *n* (%)			0.762
Low-TIL (≤10%)	45 (73.8)	257 (75.6)	
High-TIL (>10%)	16 (26.2)	83 (24.4)	
TSR, % (mean ± SD)	70.2 ± 18.1	71.2 ± 18.2	0.695
TSR group, *n* (%)			0.847
Stroma-low (TSR >50%)	52 (85.2)	293 (86.2)	
Stroma-high (TSR ≤50%)	9 (14.8)	47 (13.8)	
Ki67 LI, % (mean ± SD)	20.3 ± 12.2	10.4 ± 9.4	<0.001

RS, risk score; YAP1, Yes-associated protein 1; SD, standard deviation; TIL, tumor-infiltrating lymphocyte; TSR, tumor–stroma ratio; LI, labelling index.

**Table 3 cancers-15-05034-t003:** Logistic regression analysis for parameters associated with ODX high risk (RS ≥ 26).

	Univariate				Multivariate			
Parameters	OR	95% CI		*p*-Value	OR	95% CI		*p*-Value
		Lower	Upper			Lower	Upper	
Menopausal status				<0.001				
Premenopausal	Ref				Ref			
Menopause	2.890	1.639	5.095		2.897	1.538	5.454	0.001
YAP1 expression				<0.001				
Low	Ref				Ref			
High	0.302	0.172	0.528		0.373	0.198	0.703	0.002
Tumor size	1.018	0.720	1.441	0.918				
Histologic grade								
I	Ref							
II	2.560	0.887	7.390	0.082				
III	10.140	2.846	36.136	<0.001	4.625	1.070	19.99	0.040
Lymph node metastasis				0.002				
Absent	Ref				Ref			
Present	0.189	0.067	0.536		0.270	0.091	0.801	0.018
Lymphovascular invasion				0.051				
Absent	Ref							
Present	0.512	0.262	1.002					
TIL								
Low-TIL	Ref							
High-TIL	1.017	1.000	1.035	0.045				
TSR								
Stroma-low	Ref							
Stroma-high	0.997	0.982	1.012	0.695				
Ki67 LI	1.078	1.05	1.107	<0.001	1.062	1.035	1.09	<0.001

OR, odds ratio; CI, confidence interval; Ref, reference; YAP1, Yes-associated protein 1; TIL, tumor-infiltrating lymphocyte; TSR, tumor–stroma ratio; LI, labelling index.

**Table 4 cancers-15-05034-t004:** Parameters correlated with ODX RS.

Univariate Analysis	Beta	SE	95% CI		*p*-Value
			Lower	Upper	
Menopausal Status	2.167	0.938	0.324	4.011	0.021
Histologic grade	4.128	0.954	2.251	6.004	<0.001
Tumor size	0.592	0.581	−0.551	1.734	0.309
Lymph node metastasis	−4.064	1.045	−6.118	−2.010	<0.001
Lymphovascular invasion	−0.852	0.986	−2.791	1.087	0.388
TIL	0.091	0.033	0.027	0.156	0.006
TSR	−0.009	0.025	−0.058	0.040	0.718
Ki67 LI	0.299	0.041	0.219	0.379	<0.001
YAP1	−3.619	0.970	−5.461	−1.777	<0.001
**Multivariate Analysis**	**Beta**	**SE**	**95% CI**		* **p** * **-Value**
			**Lower**	**Upper**	
Menopausal Status	1.713	0.862	0.019	3.407	0.047
Histologic grade					
Tumor size					
Lymph node metastasis	−2.973	0.997	−4.933	−1.014	0.003
Lymphovascular invasion					
TIL					
TSR					
Ki67 LI	0.270	0.041	0.189	0.350	<0.001
YAP1	−2.816	0.894	−4.575	−1.014	0.002

SE, standard error; CI, confidence interval; TIL, tumor-infiltrating lymphocyte; TSR, tumor–stroma ratio; LI, labelling index; YAP1, Yes-associated protein 1.

## Data Availability

The datasets used and/or analyzed during the current study are available from the corresponding author on reasonable request.

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
