# Peer review of "YAP1 Expression in HR+HER2− Breast Cancer: 21-Gene Recurrence Score Analysis and Public Dataset Validation"

_cancers, 2023, doi:10.3390/cancers15205034_

Round 1

Reviewer 1 Report

Summary:

The author uses Oncotype DX 21-gene Breast Cancer Recurrence Score Assay (ODX RS) to analysis the YAP1 expression in HR+HER2 breast cancer patients versus risk factor. In their previous study, they have shown that YAP1 nuclear expression (active YAP1) is increased along with tumor stiffness. Thus, the author aimed to evaluate whether YAP1 expression correlates with ODX RS. Purpose is to use ODX RS serves as a surrogate marker for a prognosis for HR+HER2 breast cancer patients. High RS (>26) indicates that the patient belongs to the high-risk group and might benefit from chemotherapy.

Overall, the manuscript is well written and flows well. The author also indicated the limitations in their study. Although it is a surprise that role of YAP1 could be a tumor suppressor in HP+HER2 negative; base on their analysis, YAP1 could be a prognostic factor in HR+HER2- BC patients since YAP1 and ER interaction is unique in this type of patients.

Author Response

Thank you for your thoughtful and constructive feedback on our manuscript. We are pleased to know that the reviewer appreciates our work and finds the manuscript well-written. Thank you for your time and the opportunity to contribute to Cancers.

Reviewer 2 Report

GENERAL COMMENTS I was glad to review the article entitled "YAP1 expression in HR+HER2- breast cancer: 21-gene recurrence score analysis and public dataset validation". This manuscript focuses on the YAP1 relationship with the Oncotype Dx (ODX) test risk score (RS) in patients with hormone receptor-positive, and HER2-negative (HR+HER2-) breast cancer. The topic is original and relevant to the field. There is limited information on this topic in the literature. This article is well written and important as the findings of this study have the potential to significantly impact the field of oncology, providing crucial insights into the disease’s progression and aiding in the development of personalized treatment strategies for patients. In addition, the article makes clear that YAP1 may be a valuable prognostic marker and potential therapeutic target in HR+HER2- breast cancer patients. There are no further improvements regarding the methodology.

The conclusions are consistent with the evidence and arguments presented as well as summarize the main point of this article. 
References are up-to-date and appropriate
Tables and figures are well formatted and make the study easy to follow   MINOR REVISION   1) "HER2 is an established prognostic and predictive marker for patients with invasive breast cancer. The clinical and biological significance of HER2 overexpression in patients with ductal carcinoma in situ (DCIS) remains poorly defined."   Add this important information and make a brief discussion on the clinical significance of HER2 expression in DCIS Consider citing: https://pubmed.ncbi.nlm.nih.gov/36352293/   2)“In the last few years, technological developments in the medical/surgical field have been rapid and are continuously evolving. One of the most revolutionizing breakthroughs was the introduction of the IoT concept within the medical and surgical practice.”

Add this information in the discussion section and explain the role of IoT in Machine Learning related to breast cancer

Consider citing the article on the Internet of Surgical Things

https://pubmed.ncbi.nlm.nih.gov/35746359/

Author Response

Thank you for the invaluable feedback from the reviewers regarding our manuscript. We are deeply appreciative of the expertise and time you dedicated to evaluating our work.

In light of the comments, we have included a brief discussion on HER2 overexpression and DCIS, as suggested, in the discussion section (yellow highlight, line 385-391, reference 42).

However, there was a certain recommendation that we believe might not align directly with the core focus of our manuscript:

The suggestion to add content on IoT and its role in Machine Learning concerning breast cancer: Our study is primarily anchored in preclinical genetic and pathologic data. While the topic of IoT and the suggested reference about telesurgery are undoubtedly crucial in the broader scope of medical advancements, incorporating them may detract from the central theme of our manuscript.

We truly understand the value of a comprehensive discussion in research articles. Our intent is not to dismiss the suggestion but to maintain the coherence and focused narrative of our manuscript. We humbly request your understanding in this regard and are eager to receive further guidance.

Your continued insights and expertise are invaluable to us, and we are committed to refining our research for clarity and accuracy. We are grateful for your understanding and hope for the continued consideration of our manuscript for publication in Cancers.